# Knowledge of Chinese Pediatric Professionals Regarding Pediatric Pain Management

**DOI:** 10.3390/children9111785

**Published:** 2022-11-21

**Authors:** Zhen-Zhen Li, Yue-Cune Chang, Lin Gu, Jian-Fu Zhou, Bi-Rong Wei, Niang-Huei Peng

**Affiliations:** 1School of Nursing, Putian University, Putian 351100, China; 2Department of Mathematics, Tamkang University, Tamsui 25145, Taiwan; 3Neonatal Department, Affiliated Hospital of Putian University, Putian 351100, China; 4Nursing and Health School, Henan University, Kaifeng 475001, China

**Keywords:** pediatric pain management, knowledge, pediatricians, pediatric nurses, hospitalized children

## Abstract

Many healthcare professionals base their perceptions of pediatric pain on their knowledge of the subject. Therefore, knowledge deficits in this area may yield negative attitudes toward pain management and add to the complexity of pain management in hospitalized children. This study evaluated the knowledge of pediatric clinicians in China regarding pediatric pain management. Adopting a cross-sectional descriptive comparative design, we surveyed pediatric clinicians using a structured questionnaire. Inclusive criteria were pediatric clinicians, both pediatricians and nurses, with professional pediatric experience of over one year. A total of 507 pediatric clinicians participated. Most were aware of the importance of pain management in sick children but misunderstood pediatric pain, lacked knowledge for performing pediatric pain assessments and lacked knowledge for providing pain relief interventions. Background factors including differing professions (pediatricians and nurses; *p* = 0.012), age (*p* < 0.05) and hospital setting of employment (*p* = 0.003) were significantly related to clinicians’ knowledge regarding pain management. Participating pediatricians had higher levels of knowledge of pediatric pain management than nurses. Research revealed four barriers affecting clinicians’ knowledge, including misconception of pain in children, lack of professional knowledge and confidence in the practice of pediatric pain assessment, lack of professional knowledge to provide pain relief interventions, and a significant knowledge gap between pediatricians and nurses. The results point out a crucial need for multidisciplinary education to remedy these deficiencies. Further study is needed to explore strategies to strengthen clinicians’ knowledge of this vital area of practice.

## 1. Introduction

Pediatric pain management has received attention during the past decade because insufficient pain management can induce negative psychological, social and physiological effects in children and alter their quality of life [1]. Sources of pain in hospitalized children are not limited to the symptoms of illness but also include painful procedures related to physical assessment or treatment [2]. Pain in hospitalized pediatric patients can be classified as (1) neuropathic or nociceptive pain, resulting from severe injury and impacting the central somatosensory nervous system, thoracic, pelvic or abdominal viscera, and (2) acute or chronic pain, resulting from tissue injury, invasive treatment or diagnostic procedures. Although many studies advocate the importance of pain management in children, the majority of sick children continue to suffer from undertreated, moderate to severe levels of pain. Pediatric pain management can be a complex and challenging task, calling for holistic treatment administered by a multidisciplinary healthcare team [3].

In China, most previous studies have focused on children’s cancer pain and on pediatricians’ knowledge and attitudes towards pediatric cancer pain management [4], but few studies have comprehensively investigated factors impacting the knowledge of pediatric pain management of pediatric clinicians in China [5,6]. Pediatric pain management can be complicated by the effects of the developmental stage and parental influence [3,7]. In order to improve quality of life of ill children, a multidisciplinary approach is considered the best method for managing this complex condition, calling for an interdisciplinary pediatric professional team to implement such practices [8]. A discrepancy in levels of knowledge or perceptions among pediatric professionals may contribute to the inadequate pain management among hospitalized children reported by many studies [4,9]. Moreover, since many healthcare professionals base their perceptions of pediatric pain on their knowledge of the subject, knowledge deficits in this area of practice may yield negative attitudes toward pain management and add to the complexity of pain management in hospitalized children [10,11,12,13]. Not only pediatricians but also other healthcare professionals, especially nurses, play significant roles in the appropriate implementation of pediatric pain management [14,15], but few studies have analyzed the knowledge level of interdisciplinary pediatric professionals in mainland China regarding this area of practice. An urgent need exists to assess the level of knowledge regarding pediatric pain management of both pediatricians and nurses in mainland China.

The purposes of this research were to evaluate pediatric clinicians’ knowledge of pediatric pain management and to compare the knowledge levels of pediatricians and nurses working in various hospital settings in mainland China.

## 2. Research Methods

This cross-sectional descriptive comparative study surveyed pediatric clinicians using a structured questionnaire. Research was conducted over a three-month period from December 2019 to February 2020. Inclusive criteria were pediatric clinicians, both pediatricians and nurses, with professional pediatric experience of over one year who voluntarily participated in this study. Research setting included five children’s hospitals and five medical centers in Fujian Province of China. In Fujian Province, the five children’s hospitals specialized in treating pediatric patients also included departments of obstetrics and gynecology. In contrast, the medical centers in Fujian Province served various age groups of patients, not only children. Theoretically, a children’s hospital may be better equipped than other hospitals for treating ill children, but medical staff of both types of hospitals should possess similar professional qualifications. However, medical centers have longer history and richer resources than children’s hospitals.

### 2.1. Data Collection

Data were collected using either a paper-based or a web-based questionnaire. The paper-based questionnaires were distributed to participating pediatric clinicians in the research city recruited via a convenience sampling method. The participants in other cities were recruited via snowball sampling, and data were collected using a web-based survey.

### 2.2. Research Instrument

A structured questionnaire assessed pediatric clinicians’ knowledge regarding pediatric pain management. The content of the questionnaire was adapted from a previous study [13] and modified to fit the reality of clinical practice in mainland China [5,11]. The questionnaire included 9 demographic questions, and 31 items evaluating knowledge of pediatric pain management. The response options for each question were presented as a 5-point Likert-type scale as follows: 1 = strong disagreement, 2 = disagreement, 3 = unsure, 4 = agreement and 5 = strong agreement.

Research questions were divided into three sections: (1) eight questions on the recognition of pediatric pain (Q1–Q6 and Q11–Q12; four negatively worded questions, i.e., Q4, Q5, Q11 and Q12), (2) eleven questions on pediatric pain assessment (Q7, Q10 and Q13–Q21; six negatively worded questions, i.e., Q7, Q16, Q17, Q18, Q19 and Q21) and (3) twelve questions on pediatric pain management (Q8–Q9 and Q22–Q31; seven negatively worded questions, i.e., Q9, Q22, Q24, Q25, Q27, Q28 and Q31). There is a table shows the text of these questions.

### 2.3. Reliability and Validity of Research Questionnaire

In the previous study, Cronbach’s alpha coefficients of the original questionnaire were 0.86 for the total scale, 0.84 for knowledge assessment and 0.8 for attitude assessment. After the modification of the questionnaire, Cronbach’s alpha coefficient was 0.7 for the total scale. The questionnaire was reviewed by an expert panel composed of two physicians, two pediatric and neonatal nursing experts and one pediatric professor. The overall content validity index (CVI) was 0.8.

### 2.4. Statistical Analysis

Data were analyzed using Statistical Package for Social Sciences (SPSS) software, version 26 (IBM company, Taiwan). For descriptive statistics, frequencies and percentages were used for categorical data, and means and standard deviations were used for continuous data. T-tests and Chi-square tests were used to compare demographic data, and continuous and categorical data, respectively, between physicians and nurses. For knowledge scores, the descriptive statistics were means and standard deviations (SDs). The Mann–Whitney U tests were used to test the difference in the responses to each question between neonatologists and nurses.

Factors affecting the knowledge of participants regarding pediatric pain management were studied using a multiple linear regression model, with a *p*-value of less than 0.05 being considered as statistically significant.

### 2.5. Ethical Considerations

The study was approved by the Human Research Ethics Committee of the researchers’ employing university. Questionnaires were anonymous, and the participants were assured of confidentiality. Participants signed the consent form before completing the questionnaire. In other cities of Fujian Province, interested clinicians were invited to sign a web-based consent form and complete the web-based survey.

## 3. Research Results

Research data were collected from December 2019 to February 2020. Participants were 507 pediatric clinicians, including 214 physicians and 293 nurses recruited from five hospitals (*n* = 271, 53.4%) and five medical centers (*n* = 236, 45.55%) in Fujian Province in China. Response rates were 89% for the paper-based questionnaire and 65% for the web-based survey. However, all participants provided complete information; there were no missing data in the collected data.

### 3.1. Analysis of Demographic Data

The average age of the participants was 31.4 years old, with a range of 20–58 (standard deviation, 5.161) and a significant difference in the mean ages of physicians and nurses (*p* < 0.001); the mean age of physicians was 32.86 years old, and the mean age of nurses was 30.3 years old. Participating physicians had significantly longer professional pediatric experience than nurses (*p* < 0.001). No significant demographic differences appeared between the group who completed the paper questionnaire and those who completed the web questionnaire. Participant demographics are shown in Table 1.

### 3.2. Clinicians’ Knowledge Scores

The total mean knowledge score was 103.69 (SD, 8.93). The mean knowledge score of pediatricians was 104.82 (SD, 10.3), and that of pediatric nurses was 102.86 (SD, 7.70). Table 2 shows the detailed distribution and comparison of knowledge scores.

(1)Recognition of Pain in Pediatric Patients (Table 2, Q1–Q6 and Q11–Q12)

The majority of participants agreed with the statements “pain relief is a patient’s right” (Q1), “different people have different pain responses to similar stimuli” (Q2) and “children are more sensitive to painful stimuli than adults are” (Q3). There were significantly different responses between pediatricians and nurses to the above three questions (Q1–Q3) with pediatricians scoring higher than nurses. Moreover, pediatricians showed more agreement with the statement in Question 6 than nurses (*p* < 0.001).

(2)Pediatric Pain Assessment (Table 2, Q7, Q10, and Q13–Q21)

Most participants responded “unsure” to the questions regarding pediatric pain assessment (Q15–Q21). The research results found that pediatricians and nurses gave significantly different responses to questions 10, 13, 14, 15, 19, 20 and 21 (Table 2), with pediatricians scoring higher than nurses.

(3)Pediatric Pain Management Interventions (Q8–Q9 and Q22–Q31)

Twelve questions assessed the participants’ knowledge of nonpharmacological and pharmacological pain management options. The majority of participants agreed that distraction can decrease patients’ perception of pain (Q8; mean, 4.06); moreover, some nurses approved of substituting distraction interventions with pain relief medicine for crying children, while most pediatricians disapproved of this practice (Q9; *p* = 0.0018).

Regarding pharmacological pain management, most participants were unsure whether it is preferable to provide narcotics on a regular schedule or a prn schedule (as-needed range orders) for chronic pain management in children, and a significant interprofessional difference appeared between pediatricians and nurses (Q23; *p* < 0.001). Additionally, while most nurses disapproved of combining pain relief medications to manage pain, most pediatricians were unsure about this practice (Q31; *p* = 0.011; mean physicians, 3.25, nurses, 2.96).

### 3.3. Factors Impacting Knowledge Scores

Multiple linear regression models were used to compare knowledge of pediatric pain management between pediatricians and nurses, after adjusting for the effects of those potential confounding variables identified in Table 1 (*p*-value < 0.05), namely, age, gender, professional degree, department of employment, experience in pediatric profession and hospital employment. After removing those non-significant terms from the fitted model, the results were obtained, as shown in Table 3. Compared with pediatricians, nurses scored significantly lower scores, by on average 2.091 points, in knowledge of pediatric pain management (*p* = 0.012, 95% CI = −3.722 to −0.460,), after adjusting for the effects of age and hospital setting. Similarly, as shown in Table 3, participants working in children’s hospitals had significantly higher scores, by on average 2.415 points, in knowledge regarding pediatric pain management than participants from medical centers (*p* = 0.003, 95% CI = 0.810 to 4.020,), after adjusting for the effects of age and occupation. A significantly positive relationship was identified between age and knowledge scores after adjusting for the effects of occupation and hospital setting. As shown in Table 3, a 1-year age increase corresponded to, on average, a 0.311-point significant increase in knowledge scores (*p*-value < 0.001, 95% CI = 0.185 to 0.437).

### 3.4. Profession-Related Barriers to Pain Management

Because pediatric pain is difficult to assess, a clinician’s lack of knowledge in this area can be a significant barrier to appropriate treatment. Research identified several such barriers, including misconception of pediatric pain, lack of knowledge (thus, confidence in assessing and applying interventions for pediatric pain) and a gap in knowledge between physicians and nurses.

## 4. Discussion

The current research study provides an overview of the knowledge of pediatric physicians and nurses regarding pediatric pain management in Fujian Province in China. Fujian Province lies on the coast of mainland China. Its economic development is far better than that of inland provinces. The quality of healthcare and professional training for participating clinicians is comparable to that in most provinces in China.

### 4.1. Recognition of Pain in Pediatric Patients

Although most were aware of the importance of pain management in sick children (Table 1; Q1–Q3), they were less aware of the negative outcomes of poor pain management in young children (Table 1; Q5–Q6).

### 4.2. Knowledge of Pediatric Pain Assessment

Participating pediatricians displayed better knowledge of pain assessment than nurses. These results are likely consistent with previous studies in Taiwan [13]. Most participants’ responses indicated that they were unaware of the importance of pain assessment and unable to recognize pediatric patients’ pain, implying that they may improperly assess pain in ill children (Table 2, Q20). One previous study in China [16] also found similar barriers to pediatric pain management by interviewing the supervisors of pediatric surgical practices in China. The research results indicated that pediatric pain assessment may be unaddressed by pediatric clinicians in clinical practice. Insufficient knowledge of clinicians may cause them to underestimate pain assessment in their practice. Additionally, pain assessment is not a routine nursing intervention in most of pediatric wards in China; this lacking policy may be the reason why pediatric clinicians underestimate the importance of pain assessment [16]. Previous studies in other countries have found similar results and noted the importance of education in pediatric nurses to improve pediatric pain management [9,17,18].

### 4.3. Pain Management Interventions

Regarding pain management techniques, despite pediatricians’ tendency to more strongly support the practice of combining pain relief medications with respect to nurses, most participating physicians replied “unsure” regarding the effectiveness of this practice (Table 2, Q31). Based on evidence, adjuvant analgesics used in combination with other analgesics can improve the effectiveness of pain relief and reduce side effects; this method has proven useful in treating chronic pain even in children [1,7,19]. In order to improve the quality of pediatric care, there is an urgent need to advance the knowledge of pediatric clinicians regarding pain assessment and to encourage them to provide appropriate pharmacological pain relief interventions for server pain in children.

In this research, pediatricians had better knowledge of pharmacological pediatric pain management than nurses. Evidence has shown that chronic pain in children is more effectively managed with the administration of narcotics on a regular schedule than on a prn schedule [7]. However, most participants in this research were unaware of this finding. Zhang’s study found similar results in China [20].

Unfortunately, most participating nurses believed that distraction activities may be used to replace pain relief medicine for children who cry often (Table 2; Q9), indicating that they underestimated the importance of pharmacologic pain relief and overestimated the effectiveness of distraction in pediatric pain control. Moreover, the responses of nurses to this question implied that they did not believe that crying in children was closely related to painful stimuli. Nurses’ lack of awareness of the negative outcomes of pain in children and lack of knowledge of pediatric pain assessment may have impacted their response to this question.

Children who suffer from acute pain or severe chronic pain may require prescribed analgesic medicine and a combination of pharmacological medicines, such as analgesic and sedative drugs, to relieve pain. Distraction may be effective in children but often works as a complementary treatment for the relief of anxiety or fear during invasive medical interventions or shifting attention away from chronic pain.

### 4.4. Factors Impacting Knowledge Scores

Based on the multiple linear regression analysis, the professional position (nurses vs. pediatricians) and the employment setting (children’s hospitals vs. medical centers) significantly impacted knowledge of pediatric pain management (Table 3). Age also significantly impacted knowledge scores. The influence of the age of participants could be explained by the fact that participating physicians were on average older than participating nurses.

Additionally, the results indicated that most pediatric clinicians lacked the knowledge of pain assessment and how to provide interventions in very young children, such as neonates. Previous studies also found improper pain assessment and inadequate staff knowledge to be the most frequently cited barriers to pain management [11,14,15,17]. A previous qualitative research study also found that Chinese pediatric managers underestimated the negative effects of painful procedures on ill neonates and were especially unaware that invasive interventions could cause neonatal injury or crying [16].

In this research, physicians showed better knowledge of pediatric pain management than nurses, possibly as the result of published guidelines for pediatricians in China regarding pediatric pain management and recent pediatric cancer pain management training for pediatricians in medical schools and resident education [5,6]. In contrast, in China, pediatric nurses lack the opportunity to receive such training, which may explain these results [5,11]. Moreover, in China, the prescription of opioid and sedation drugs is the physicians’ responsibility, while nurses are responsible to perform accurate pain assessments, refer the results of assessment to physicians, administer medication according to physicians’ orders, provide nonpharmacological pain relief measures, and evaluate the effects of pain management [5,6]. Pediatric nurses are members of the treatment team who are constantly close to the hospitalized patient and may play a role as the first decision maker in pain management; their pain assessments influence medical judgment and patients’ perception of pain control [11,18].

These findings suggest an urgent need to improve the knowledge of all pediatric clinicians, especially nurses, regarding pain assessment and pain management interventions in order to close the existing knowledge gap and provide holistic pain management for ill children and comfort for their families [9,12]. In order to effectively promote pediatric clinicians’ knowledge regarding pediatric pain assessment and pain management interventions, we suggest that pain management policy and practice guidelines be established to guide clinical practice and that regular continuing staff training be offered to empower pediatric clinicians to provide sufficient pain management. Moreover, there is an urgent need for continuing the study of existing barriers and potential strategies to advance pediatric pain management in China.

## 5. Study Limitations

Research data were obtained from one province in China, implying an inability to generalize the findings. Additionally, results based on self-reporting may be exaggerated in reported frequencies and may not correspond to actual clinical practice. In spite of these two limitations, the present study may be the first to compare physicians and nurses in China on the basis of knowledge of pediatric pain management. Because quality pain management requires a multidisciplinary teamwork approach, assessing the present knowledge of pediatric clinicians is an important first step in strengthening medical teams and improving clinical practice.

## 6. Conclusions

Research revealed four barriers related to the knowledge of clinicians, including misconceptions regarding pain in children, lack of professional knowledge and confidence to practice pediatric pain assessment, lack of professional knowledge to provide pain management interventions, and a significant knowledge gap between pediatricians and nurses. Further studies should explore strategies to implement multidisciplinary pain management education to boost pediatric clinicians’ professional knowledge and to close the professional knowledge gap between pediatricians and nurses.

## Figures and Tables

**Table 1 children-09-01785-t001:** Demographic profile of participants (N = 507; 214 physicians and 293 nurses).

Characteristic	N (%)	Physicians, Number (%)	Nurses, Number (%)	T/X^2^	*p*-Value
Age (years), mean ± SD		32.86 ± 8.13	30.30 ± 5.54	3.978	<0.001 ^a^
Gender				200.698	<0.001 ^b^
Female	388 (76.53%)	97 (45.3%)	291 (99.3%)		
Male	119 (23.47%)	117 (54.7%)	2 (0.7%)		
Marital status				0.285	0.593 ^b^
Married	343 (67.65%)	142 (66.4%)	201 (68.6%)		
Unmarried	164 (32.35%)	72 (33.6%)	92 (31.4%)		
Parenting experience				0.353	0.838 ^b^
None	199 (39.25%)	83 (38.8%)	116 (39.6%)		
One child	182 (35.90%)	75 (35.0%)	107 (36.5%)		
Two or more children	126 (24.85%)	56 (26.2%)	70 (23.9%)		
Professional Degree				176.987	<0.001 ^b^
Associate Bachelor	132 (26.04%)	0 (0.00%)	132 (45.1%)		
Bachelor’s degree	322 (63.51%)	161 (75.2%)	161 (54.9%)		
Master’s degree	53 (10.45%)	53 (24.8%)	0 (0.00%)		
Professional Position				0.312	0.577 ^b^
Clinical staff	484 (95.46%)	203 (94.9%)	281 (95.9%)		
Managers	23 (4.54%)	11 (5.1%)	12 (4.1%)		
Department of Employment					
NICU *	226 (44.58%)	63 (29.4%)	163 (55.6%)		
PICU **	97 (19.13%)	42 (19.6%)	55 (18.8%)		
Pediatric wards	184 (36.29%)	109 (51.0%)	75 (25.6%)		
Experience in Pediatric Profession				22.479	<0.001 ^b^
1–5 years	206 (40.63%)	101 (47.2%)	105 (35.8%)		
6–10 years	152 (29.98%)	40 (18.7%)	112 (38.2%)		
Over 10 years	149 (29.39%)	73 (34.1%)	76 (25.9%)		
Hospital of Employment				45.425	<0.001 ^b^
Children’s hospital	271 (53.45%)	77 (36.0%)	194 (66.2%)		
Medical center	236 (46.55%)	137 (64.0%)	99 (33.8%)		

^a^, *t*-test; ^b^, chi-square tests. * NICU, neonatal intensive care units. ** PICU, Pediatric Intensive Care Units.

**Table 2 children-09-01785-t002:** Comparison of knowledge scores using Mann–Whitney U tests (N = 507; 214 physicians and 293 nurses).

Characteristic	Pediatricians, Mean ± SD	Nurses, Mean ± SD	z	*p*-Value
1. It is a patient’s right to expect total pain relief as a consequence of treatment.	4.48 ± 0.669	4.18 ± 0.625	−5.620	<0.001 *
2. Comparable stimuli in different children may produce different pain responses.	4.58 ± 0.573	4.41 ± 0.625	−3.421	0.001 *
3. Children are more sensitive to painful stimuli than are adults.	4.20 ± 0.776	3.90 ± 0.874	−4.000	<0.001 *
4. Sedation is an effective method of pain relief for neonates. (negative)	2.71 ± 1.089	2.72 ± 0.835	−0.177	0.860
5. Pain in the neonatal period has no negative effects on growth and development. (negative)	3.19 ± 1.220	3.15 ± 0.880	−1.147	0.251
6. Neonates are more likely to experience long-term consequences from painful experiences than are older children.	3.82 ± 0.904	3.43 ± 0.811	−5.274	<0.001 *
7. It is unnecessary to provide medication or pain relief interventions when an ill infant does not cry or twist his/her body during invasive procedures. (negative)	3.30 ± 1.188	3.17 ± 1.031	−1.751	0.080
8. Distraction (using music or relaxation techniques) can decrease the perception of pain or symptoms of discomfort.	4.10 ± 0.619	4.03 ± 0.542	−1.361	0.174
9. Distraction activities may be used to replace pain relief medication for children who cry often. (negative)	2.78 ± 1.081	4.03 ± 0.957	−2.372	0.018 *
10. Infants/children/adolescents may sleep in spite of severe pain.	3.85 ± 0.888	3.72 ± 0.817	−2.034	0.042 *
11. Parents should not be present during painful procedures. (negative)	2.82 ± 1.218	2.97 ± 1.146	−1.345	0.179
12. Children with pain should be encouraged to endure as much pain as possible before resorting to a pain relief measure. (negative)	2.62 ± 1.076	2.89 ± 1.044	−2.771	0.006 *
13. For effective treatment of pain, it is necessary to continuously assess the pain and efficacy of the therapy.	4.31 ± 0.619	4.20 ± 0.600	−2.149	0.032 *
14. Lack of pain expression in children does not mean lack of pain.	4.27 ± 0.714	4.09 ± 0.669	−3.570	<0.001 *
15.The most accurate judge of the intensity of pediatric patients’ pain is the patient themselves.	3.93 ± 0.932	3.78 ± 0.831	−2.281	0.023 *
16. The only way to determine the discomfort or pain of infants is to assess their behaviors. (negative)	2.87 ± 1.147	2.88 ± 1.027	−0.105	0.916
17. Estimation of pain by an M.D. or R.N. is as valid a measure of pain as a sick child’s self-report. (negative)	2.56 ± 0.985	2.58 ± 0.843	−0.269	0.788
18. Children younger than preschool age cannot reliably report their pain intensity. Therefore, nurses should rely on the parents’ assessment of a child’s pain intensity. (negative)	3.07 ± 1.120	3.27 ± 0.962	−1.832	0.067
19. When a pediatric patient complains of severe pain his/her vital signs should be evaluated. (negative)	1.97 ± 0.750	2.32 ± 0.832	−5.096	<0.001 *
20. I have confidence in my ability to recognize patient’s pain by assessing their vital signs and behaviors.	3.48 ± 0.892	3.24 ± 0.784	−3.245	0.001 *
21. Placebo medication may be used to assess pain in children. (negative)	2.59 ± 0.934	2.78 ± 0.717	−2.271	0.023 *
22. Because narcotics may cause respiratory depression, they should not be used in pediatric patients. (negative)	2.69 ± 1.190	2.57 ± 0.982	−1.074	0.283
23. For chronic pain, giving narcotics on a regular schedule is better than on a “prn” schedule. (prn: as needed range orders for medicine)	3.79 ± 0.880	3.41 ± 0.877	−4.887	<0.001 *
24. Pediatric patients should show discomfort before receiving the next dose of pain relief medication.	3.21 ± 1.101	3.30 ± 0.954	−0.317	0.751
25. Intramuscular injection is the best way to provide pain relief medication.	3.24 ± 1.156	3.26 ± 0.847	−0.989	0.323
26. After the initial recommended dose of opioid analgesic, subsequent doses should be adjusted in accordance with the individual patient’s response.	4.23 ± 0.653	3.93 ± 0.590	−5.850	<0.001
27. Increasing analgesic requirements indicate that the pediatric patient is becoming addicted to the narcotic.	2.87 ± 1.012	2.86 ± 0.823	−0.655	0.513
28. When a pediatric patient or his/her parents request increasing amounts of analgesics for pain control, this usually indicates that the patient may be physiologically dependent.	2.77 ± 1.061	2.80 ± 0.858	−0.007	0.995
29. Pediatric patients having severe chronic pain often need higher dosages of pain meds than pediatric patients with acute pain.	3.39 ± 1.032	3.22 ± 0.910	−1.871	0.061
30. Appropriate dosage of narcotics depends on the effectiveness of pain management for the patients rather than on the maximum dosage as determined by body weight.	4.21 ± 0.690	4.00 ± 0.654	−3.847	<0.001
31. When the effectiveness of pain relief medicine is poor, it should not be combined with another pain relief medicine. (negative)	2.96 ± 1.160	3.25 ± 0.876	−2.544	0.011 *

Note: Coding: 1 = strong disagreement, 2 = disagreement, 3 = unsure, 4 = agreement and 5 = strong agreement. * *p*-value < 0.05.

**Table 3 children-09-01785-t003:** Factors impacting knowledge scores using multiple linear regression (N = 507).

Variable	B	SE	95% Wald CI	Wald X^2^	*p*-Value
Intercept	87.766	2.867	(82.147, 93.385)	937.217	0.000
Nurses vs. pediatricians ^a^	−2.091	0.832	(−3.722, −0.460)	6.311	0.012 *
Age	0.311	0.064	(0.185, 0.437)	23.304	<0.001 *
Children’s hospital vs. medical center ^a^	2.415	0.819	(0.810, 4.020)	8.701	0.003 *

Note: Dependent variable: knowledge scores. Model: (intercept) occupation, age and hospitals. ^a^: reference group. Abbreviations: SE, standard error; CI, confidence interval; vs., versus. * *p*-value < 0.05.

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
