# Peer review of "Knowledge of Chinese Pediatric Professionals Regarding Pediatric Pain Management"

_children, 2022, doi:10.3390/children9111785_

Round 1

Reviewer 1 Report

Thank you for the opportunity to review this manuscript, entitled Knowledge of Chinese Pediatric Professionals Regarding Pediatric Pain Management. This manuscript describes a study of pediatric healthcare professionals, including physicians and nurses, and differences in their knowledge regarding pain assessment and management in pediatric patients in five children’s hospitals and five medical centers in the Fujian Province of China. Overall, this paper represents an important contribution to the literature, as few known previous studies have been conducted to evaluate differences between pediatric nurses and physicians and pain management knowledge in China. However, several suggestions to strengthen this manuscript are below.

1)      Language: Overall, this manuscript is well-written, although this reviewer experienced confusion regarding some conflicting terminology. Clarification of the terms and concepts below could strengthen this manuscript.

a.       “Knowledge” and other concepts like “confidence”: As written, it appears that the only concept that was evaluated through the research instrument was “pediatric clinicians’ knowledge regarding pediatric pain management.” However, it appears that throughout the paper, the term “knowledge” is used interchangeably with other related but distinct terms like “confidence” (for example, page 8 line 205), “skill” (for example, page 8 line 224), and “communication” (for example, page 9 line 263). Since only knowledge was evaluated in this study, conclusions about professionals’ skills or confidence cannot be extrapolated from these results.

b.       “Pain assessment” and “pain management”: It appears that the research instrument evaluated both pain assessment and pain management, but these two terms are sometimes conflated throughout the manuscript. For example, in the title, abstract, and throughout the discussion, only “pain management” is described, even though results for both assessment and management are described in the results.

2)      Description of hospital sites in the methods:

a.       This reviewer feels that having a stronger understanding of the research sites in which this study was conducted would strengthen the interpretation of the results of this study. For example, what is the difference between “children’s hospital” and “medical center”? What is the typical pediatric census within these different settings? Are there any major differences across the 10 research sites in terms of acuity of patients, type of patients (for example, oncology hospital versus general hospital; trauma hospital), etc.? A more detailed description of the research sites could help the reader better situate the study findings.

3)      Analysis:

a.       In general, the statistical analysis section should include more details about the analysis. For example, how were missing data handled? For the multiple regression model, what covariates were included in the model and why (justification for inclusion)?  

b.       This reviewer is unclear about why certain demographic variables were included or excluded as covariates in the multivariate analysis:

                                                               i.      Why was marital status evaluated in this study and why was it treated as a covariate, given that marital status doesn’t seem to have an obvious impact on professional practice (at least not in Western countries)?

                                                             ii.      Why was age included in the model versus years of experience in the profession? It seems that years of experience should have a more direct impact on professional practice than age (such as clinicians who may have come into healthcare from another profession later in their career).

                                                           iii.      Why was unit of practice (PICU/NICU versus general pediatric ward) not further explored, as this could have strong implications for how often physicians and nurses see and have the opportunity to assess and manage pediatric pain?

                                                           iv.      Why was education type (e.g., Associate Bachelor, Bachelor’s, versus Master’s degrees) not considered as a covariate, as again, this could have a potential impact on healthcare knowledge?

4)      Stronger discussion section: In general, the discussion section could be strengthened in the following ways:  

a.       It would be interesting to understand how these research sites may differ from other sites across China. How representative was the sample in terms of pediatricians and nurses across Fujian Province (e.g., training, level of experience)? Have there been previous similar studies conducted elsewhere in China, even if in adult pain management, that could provide some points of comparison with the healthcare professionals included in this study? This may have implications for if local or regional efforts are needed to improve pediatric pain practice versus the need for cross-cutting, national efforts.

b.       The following paper may be helpful to the authors:

                                                               i.      Hu, J., Ruan, H., Li, Q., Gifford, W., Zhou, Y., Yu, L., Harrison, D. (2020). Barriers and Facilitators to Effective Procedural Pain Treatments for Pediatric Patients in the Chinese Context: A Qualitative Descriptive Study. Journal of Pediatric Nursing, 54, 78-85.

5)      Stronger conclusion section

a.       Similar to the above comment about the interchangeable use of “knowledge” with other concepts, the conclusions about confidence and communication cannot be drawn based on the study results, which focus on knowledge of pediatric pain assessment and management. Similarly, the conclusion “five barriers to pediatric pain management implementation” cannot be drawn, as the study and study instruments did not focus on the implementation of pain management, only about knowledge of pain management.

Author Response

1) Language: Overall, this manuscript is well-written, although this reviewer experienced confusion regarding some conflicting terminology. Clarification of the terms and concepts below could strengthen this manuscript.

“Knowledge” and other concepts like “confidence”: As written, it appears that the only concept that was evaluated through the research instrument was “pediatric clinicians’ knowledge regarding pediatric pain management.” However, it appears that throughout the paper, the term “knowledge” is
used interchangeably with other related but distinct terms like “confidence” (for example, page 8 line 205), “skill” (for example, page 8 line 224), and “communication” (for example, page 9 line 263). Since only knowledge was
evaluated in this study, conclusions about professionals’ skills or confidence cannot be extrapolated from these results.

Thank you for your input.  Knowledge of pain assessment, confidence, and skills are the subsystems of knowledge in pediatric pain management and were the assessed by the questions within the research questionnaire. 

“Pain assessment” and “pain management”: It appears that the research instrument evaluated both pain assessment and pain management, but these two terms are sometimes conflated throughout the manuscript. For example, in the title, abstract, and throughout the discussion, only “pain management” is described, even though results for both assessment and management are described in the results.

We renamed the sub-domain of pediatric pain management into “Pain Management Interventions”.

Description of hospital sites in the methods:

This reviewer feels that having a stronger understanding of the research sites in which this study was conducted would strengthen the interpretation of the results of this study. For example, what is the difference between “children’s hospital” and “medical center”? What is the typical pediatric census within
these different settings? Are there any major differences across the 10 research sites in terms of acuity of patients, type of patients (for example, oncology hospital versus general hospital; trauma hospital), etc.? A more detailed description of the research sites could help the reader better situate the
study findings.

Thank you. Additional explanation on hospital setting has been added to the
manuscript. 

Analysis: 

In general, the statistical analysis section should include more details about the
analysis. For example, how were missing data handled? For the multiple regression model, what covariates were included in the model and why (justification for inclusion)?

This reviewer is unclear about why certain demographic variables were included or excluded as covariates in the multivariate analysis:

i. Why was marital status evaluated in this study and why was it treated as a covariate, given that marital status doesn’t seem to have an obvious impact on professional practice (at least not in Western countries)?

ii. Why was age included in the model versus years of experience in the profession? It seems that years of experience should have a more direct impact on professional practice than age (such as clinicians who may have come into healthcare from another profession later in their career).

iii. Why was unit of practice (PICU/NICU versus general pediatric ward) not further explored, as this could have strong implications for how often physicians and nurses see and have the opportunity to assess and manage pediatric pain?

iv. Why was education type (e.g., Associate Bachelor, Bachelor’s, versus Master’s
degrees) not considered as a covariate, as again, this could have a potential impact on healthcare knowledge?

  1. Thanks for your helpful comments. We have provided more detailed
    descriptions of our analysis. In this study, the response rates were 89%
    for the paper-based questionnaire and 65% for the web-based survey.
    However, all participants provided complete information. Thus there
    were no missing data in the collected data. For the multiple regression
    model, we also provide more details descriptions.
  2. Marital status was one of the demographic data surveyed, but it did not have an impact on the professional knowledge in our research. 
  3. In this research, there was a hypothesis that was “the years of experience in profession may have a significantly impact factor on professional knowledge in clinicians.” However, there was no significant relationship (reject this hypothesis). On the other hand, there is a significant impact regarding the age of participants on the professional knowledge
    in these clinicians. 
  4. Both age and years of professional experience were included in our model. We hypothesized that “ years of experience in profession may be a significant impact factor on professional knowledge in clinicians.” However, we found no significant relationship and so we rejected this
    hypothesis. On the other hand, we did find a significant impact of age on the professional knowledge in these clinicians. 
  5. We have explored all potential factors, including the unit of practice, age,
    professional experience, and professional degree, on the level of knowledge in clinicians. However, only three significant impact factors were found.
    We do appreciate your careful reviewing of this manuscript. We rewrote the multiple regression model in this revised manuscript trying to clarify these confusions.

This reviewer feels that having a stronger understanding of the research sites in which this study was conducted would strengthen the interpretation of the results of this study. For example, what is the difference between “children’s hospital” and “medical center”? What is the typical pediatric census within these different settings? Are there any major differences across the 10 research sites in terms of acuity of patients, type of patients (for example, oncology hospital versus general hospital; trauma hospital), etc.? A more detailed description of the research sites could help the reader better situate the study findings.

A paragraph has been added to describe the research setting. 

The following paper may be helpful to the authors: Hu, J., Ruan, H., Li, Q., Gifford, W., Zhou, Y., Yu, L., Harrison, D. (2020). Barriers and Facilitators to Effective Procedural Pain Treatments for Pediatric Patients in the Chinese Context: A Qualitative Descriptive Study. Journal of Pediatric Nursing, 54, 78-
85

Thanks a lot for this helpful information. 

Stronger conclusion section 

Similar to the above comment about the interchangeable use of “knowledge” with other concepts, the conclusions about confidence and communication cannot be drawn based on the study results, which focus on knowledge of pediatric pain assessment and management. Similarly, the conclusion “five barriers to pediatric pain management implementation” cannot be
drawn, as the study and study instruments did not focus on the implementation of pain management, only about knowledge of pain management.

The conclusion section has been revised.

Reviewer 2 Report

The authors set out to assess their region's recognition and adequate treatment of childhood pain.

The survey was carried out with the help of a paper-based or electronically filled-out validated questionnaire for pediatric pain relief.

The relatively large number of participants was categorized according to gender, age, marital status, professional, employment conditions, and working years. Finally, these data were compared with the proportion of correct answers.

Different professionals (pediatricians vs. nurses; p= 0.012), age (p < 0.05) and hospital setting of employment (p =0.003) were significantly related to clinicians’ knowledge regarding pain management. Five barriers to pediatric pain management implementation were identified as follows:

The misconception of neonatal pain

Lack of both knowledge and confidence in treating pain

Difficulty communicating about pain management

A gap in knowledge between pediatricians and nurses

The article is well-organized and the sections are well-developed. The authors did a good job of synthesizing the literature. The authors answer the questions they set out to evaluate pediatric clinicians' knowledge of pediatric pain management and compare the knowledge levels of pediatricians and nurses working in various hospital settings in mainland China. The methodology is clearly explained. Data were collected by either a paper-based or a web-based questionnaire. A structured questionnaire assessed pediatric clinicians' knowledge regarding pediatric pain management: the questionnaire's content was adapted from a previous study and modified to fit the reality of clinical practice in mainland China. The questionnaire included nine demographic questions and 31 items evaluating the knowledge of pediatric pain management. The study was ethically approved. Questionnaires were anonymous, and the participants have assured confidentiality. The theory fully connects to the data. The article is well-written and easy to understand.

The author's results are convincing, although there are some questions regarding particular items.

Which part can be underlined as a universal "take home message"? A paper that encourages the education of the nurses (and doctors) in this field in your region is essential. The impact is high because children suffer from pain and the recognition and treatment are insufficient. As it is written: "…few studies have analyzed the knowledge level of interdisciplinary pediatric professionals in mainland China regarding this area of practice. There is an urgent need to assess the level of knowledge regarding pediatric pain management among both pediatricians and nurses in mainland China."

 Some additional remarks:

"Professional degree differed significantly between physicians and nurses because all physicians hold a bachelor's MD degree (p < 0.001)" - I would not write it as a result.

"In this research, pediatricians had better knowledge of pharmacological pediatric pain management than did nurses." – why is it interesting?   

In the introduction, there are four different types of pain: acute, neuropathic, visceral, and chronic. It may be better to write about nociceptive or neuropathic and acute or chronic.

 It would be helpful to know the results of the different clinical practice areas e.g., surgery, anesthesiology and emergency, oncology, and general pediatrics.

It would be helpful to clarify the differences between the Children's Hospital and Medical Center in this region.

There is a misspelling: "Further study is needed to explore strategies to strengthen clinicians' knowledge of this viral area of practice."

Author Response

"Professional degree differed significantly between physicians and nurses because all physicians hold a bachelor's MD degree (p< 0.001)" - I would not write it as a result.

This sentence has been removed. 

"In this research, pediatricians had better knowledge of pharmacological pediatric pain management than did nurses." – why is it interesting? 

Yes, it is to be expected that pediatricians had more knowledge than nurse did.
However, since nurses are responsible to give prescribed medication to patients, they should also know why and how to administer medication.

In the introduction, there are four different types of pain: acute, neuropathic, visceral, and chronic. It may be better to write about nociceptive or neuropathic and acute or chronic.

This has been revised. 

It would be helpful to know the results of the different clinical practice areas e.g.,
surgery, anesthesiology and emergency, oncology, and general pediatrics.

Yes, it would be interesting. I will keep this suggestion in mind for further research. 

It would be helpful to clarify the differences between the Children's Hospital and
Medical Center in this region.

This information has been added. 

There is a misspelling: "Further study is needed to explore strategies to strengthen clinicians' knowledge of this viral area of practice."

This misspelling has been cored.

Reviewer 3 Report

Thank you for the opportunity to review the manuscript entitled "Knowledge of Chinese Pediatric Professionals Regarding Pediatric Pain Management". Overall, it was well written and a very novel approach to understanding knowledge deficits for pediatric pain management. The manuscript would benefit from a few minor changes, points of clarification, and addition to the discussion. See comments below.

Introduction:  There is a focus on hospitalized children experiencing pain, but a large number of youth who attend outpatient appointments with pediatricians are also experiencing pain that is not well controlled. Please expand the introduction to decrease focus on hospitalized youth only.

Methods: The authors commented on the adaption of the measure to "fit the reality" of the clinical practice in China. Please provide more details on this so readers know what this adaption entailed.

Discussion: The discussion largely focuses on differences between nurses and physicians responses. However, the most interest aspects of the results are highlighted in 2-3 sentences in the conclusion section only. Please expand on what can be done to improve knowledge. 

Author Response

Thank you for the opportunity to review the manuscript entitled
"Knowledge of Chinese Pediatric Professionals Regarding Pediatric Pain Management". Overall, it was well written and a very novel approach to understanding knowledge deficits for pediatric pain management. The manuscript would benefit from a few minor changes, points of clarification, and addition to the discussion. See comments below.

Introduction: There is a focus on hospitalized children experiencing pain, but
a large number of youth who attend outpatient appointments with pediatricians
are also experiencing pain that is not well controlled. Please expand the introduction to decrease focus on hospitalized youth only.

Thank you. This has been revised to refer not only to hospitalized children.

Methods: The authors commented on the adaption of the measure to "fit the reality" of the clinical practice in China. Please provide more details on this so readers know what this adaption entailed.

ELMA cream is not often sued in clinical practice in mainland China. In mainland
China, pain assessment is not regular nursing routine in most hospitals. This is the policy issue. I have mentioned the condition on the manuscript. 

Discussion: The discussion largely focuses on differences between nurses and
physicians responses. However, the most interest aspects of the results are
highlighted in 2-3 sentences in the conclusion section only. Please expand on
what can be done to improve knowledge.

Thank you. More recommendations are given at the end of Section 4, Factors Impacting Knowledge Scores, as potential methods of remedying the knowledge gaps discussed.

Round 2

Reviewer 1 Report

Thank you for the opportunity to re-review this paper. The authors have adequately addressed my prior concerns. My one remaining comment is that while the survey evaluated knowledge and included an item about confidence, this reviewer still doesn't believe the survey as described could evaluate skills. Evaluating skills would require hands-on observation and assessment of the professional performing clinical competencies, whether in clinical practice or in a simulated environment. I would still suggest that the authors remove "skill" from the findings and discussion. Otherwise, I have no further comment. 

Author Response

Thank you for your comments, we removed the "skill" from the sections of outcomes and discussion. 

Reviewer 2 Report

All but one answer are acceptable.

In the introduction, there are four different types of pain: acute, neuropathic, visceral, and chronic. It may be better to write about nociceptive or neuropathic and acute or chronic.

This part should still be revised. 

Author Response

For your concern, we agreed with your suggestion and revised that.